# Comparison of anterior nares C$_T$ values in asymptomatic and symptomatic individuals diagnosed with SARS-CoV-2 in a university screening program

Samantha M. Hall[1], Lena Landaverde[2], Christopher J. Gill[3], Grace M. Yee[4], Madison Sullivan[5], Lynn Doucette-Stamm[6], Hannah Landsberg[5], Judy T. Platt[5], Laura White[7], Davidson H. Hamer[3,8,9,10], Catherine M. Klapperich[2,6,10]*

1 Department of Environmental Health, Boston University School of Public Health, Boston, Massachusetts, United States of America, 2 Department of Biomedical Engineering, Boston University, Boston, Massachusetts, United States of America, 3 Department of Global Health, Boston University School of Public Health, Boston, Massachusetts, United States of America, 4 Occupational Health Center, Boston University, Boston, Massachusetts, United States of America, 5 Student Health Services, Healthway, Boston University, Boston, Massachusetts, United States of America, 6 Clinical Testing Laboratory, Boston University, Boston, Massachusetts, United States of America, 7 Department of Biostatistics, School of Public Health, Boston University, Boston, Massachusetts, United States of America, 8 National Emerging Infectious Diseases Laboratory, Boston University, Boston, Massachusetts, United States of America, 9 Center for Emerging Infectious Disease Research and Policy, Boston University, Boston, MA, United States of America, 10 Precision Diagnostics Center, Boston University, Boston, MA, United States of America

* catherin@bu.edu

**Data Availability Statement:** Data set has been uploaded to Figshare with DOI 10.6084/m9.figshare.20134124.

## Abstract

At our university based high throughput screening program, we test all members of our community weekly using RT-qPCR. RT-qPCR cycle threshold (C$_T$) values are inversely proportional to the amount of viral RNA in a sample and are a proxy for viral load. We hypothesized that C$_T$ values would be higher, and thus the viral loads at the time of diagnosis would be lower, in individuals who were infected with the virus but remained asymptomatic throughout the course of the infection. We collected the N1 and N2 target gene C$_T$ values from 1633 SARS-CoV-2 positive RT-qPCR tests of individuals sampled between August 7, 2020, and March 18, 2021, at the BU Clinical Testing Laboratory. We matched this data with symptom reporting data from our clinical team. We found that asymptomatic patients had C$_T$ values significantly higher than symptomatic individuals on the day of diagnosis. Symptoms were followed by the clinical team for 10 days post the first positive test. Within the entire population, 78.1% experienced at least one symptom during surveillance by the clinical team (n = 1276/1633). Of those experiencing symptoms, the most common symptoms were nasal congestion (73%, n = 932/1276), cough (60.0%, n = 761/1276), fatigue (59.0%, n = 753/1276), and sore throat (53.1%, n = 678/1276). The least common symptoms were diarrhea (12.5%, n = 160/1276), dyspnea on exertion (DOE) (6.9%, n = 88/1276), foot or skin changes (including rash) (4.2%, n = 53/1276), and vomiting (2.1%, n = 27/1276). Presymptomatic individuals, those who were not symptomatic on the day of diagnosis but became symptomatic over the following 10 days, had C$_T$ values higher for both N1 (median = 27.1, IQR 20.2–32.9) and N2 (median = 26.6, IQR 20.1–32.8) than the symptomatic group N1

**Funding:** The work here was funded by Boston University as part of the Back to BU COVID-19 surveillance testing program. The funders had no role in study design, data analysis, decision to publish, or preparation of the manuscript. Some demographic data was provided from BU administrative entities.

**Competing interests:** The competing interests for Drs. Connor and Klapperich listed in the original submission do not alter our adherence to PLOS ONE policies on sharing data and materials.

(median = 21.8, IQR 17.2–29.4) and N2 (median = 21.4, IQR 17.3–28.9) but lower than the asymptomatic group N1 (median = 29.9, IQR 23.6–35.5) and N2 (median = 30.0, IQR 23.1–35.7). This study supports the hypothesis that viral load in the anterior nares on the day of diagnosis is a measure of disease intensity at that time.

## Introduction

Reverse-transcription polymerase chain reaction (RT-qPCR) is the most widely used test for detecting SARS-CoV-2 infection through RNA detection [1, 2]. Optimized RT-qPCR protocols report cycle threshold ($C_T$) values for primer-specific viral antigens such as the nucleocapsid antigens (N1 and N2) along with the human housekeeping gene RNA polymerase (Rnase P), which is used as a control demonstrating that the swab made effective contact with the nasopharyngeal mucosa [3]. $C_T$ values are inversely proportional to the viral load and are therefore a relative measure of infectivity. Since $C_T$ reflects viral load, and viral load is an important predictor of disease severity, it is logical to assume that these values would differ between individuals with symptomatic vs. asymptomatic SARS-CoV-2 infection. To test that assumption, we analyzed data generated through a comprehensive and systematic testing system set up at a large urban university [4]. As asymptomatic and pre-symptomatic transmission of SARS-CoV-2 have contributed to the devastating burden of this disease [5], information relating $C_T$ values, timing of symptom onset and viral load, and transmissibility potential by viral load may hold important implications for continued COVID-19 mitigation efforts [6–8].

This analysis combines clinical symptom evaluation, epidemiological contact tracing data, and laboratory investigations into $C_T$ values for 1633 SARS-CoV-2 positive individuals who were sampled between August 7, 2020, and March 18, 2021, and tested at the Boston University (BU) Clinical Testing Laboratory. In this analysis, we sought to understand the relationship between PCR signal intensity at the time of initial identification and the symptom status of those individuals across their arc of infection. Our hypothesis was that PCR signal intensity, being a measure of viral load and hence a marker for infection intensity, would be higher among symptomatic individuals compared with those who remained asymptomatic.

## Materials and methods

This retrospective analysis details disease progression for all students and employees who tested positive through the BU Clinical Testing Laboratory from the start of the testing program on August 7, 2020, through March 18, 2021. All cases occurred prior to the introduction of the Delta and Omicron variants into our population based on contemporaneous sequencing of isolates. All sequenced isolates from Boston University have been uploaded to the GISAID database (https://www.gisaid.org/). Data collection for this analysis was a concerted effort between clinical professionals, BU's contact tracing team, laboratory workers, and public health professionals as part of BU's "Back 2 BU" effort. BU Healthway and the Back 2 BU initiative have been detailed previously [4, 9].

### Data collection

Data on positive cases of SARS-CoV-2 were collected from BU electronic medical records (EMRs). Positive cases were excluded if: the test was positive within the 90 days after initial BU positive, and therefore likely due to residual viral shedding from initial SARS-CoV-2 infection;

attributed to suspected amplicon contamination in research laboratory settings [10]; determined to be a false positive by the clinical team; or transferred out of the BU screening program before completion of 10 day isolation, as in a student was no longer enrolled or an employee no longer worked for BU. Our cohort included individuals working in laboratories that were using amplified genetic material from SARS-CoV-2. People who tested positive and were not truly infected were identified by sequencing and clinical determination and were excluded from this study [10].

Demographic and study variables were collected for each positive individual. Symptom data, including initial symptom onset and type of symptoms experienced for symptomatic individuals, were collected from the notes of clinical staff for the extent of an individual's BU-monitored isolation period. Individuals were classified as follows: *Presymptomatic* patients were defined as people who first experienced symptoms on day zero of the positive test or any day following a positive test up to ten days. *Asymptomatic* patients were considered to have developed no COVID-19 related symptoms for the time prior to or in the ten days following a positive test. *Symptomatic* patients showed symptoms on or before the day of testing as recorded by a daily attestation questionnaire.

## Sample collection and analysis

RT-qPCR tests were used to identify presence of SARS-CoV-2 RNA in self-administered anterior nares swabs. The testing pool during the time period of this study included students and employees who came to campus for any reason. All undergraduates were tested twice a week, and graduate students and employees were tested once a week while working on campus. The total campus population during this time was approximately 40,000 people. The SARS-CoV-2 RT-qPCR assay was based on the CDC primer set and was optimized by the BU Clinical Testing Laboratory [11]. Validation data was sent to the FDA in an Emergency Use Authorization application in July 2020 [9]. Only RT-qPCR tests processed by BU were used in this study; select documented nucleic acid tests from non-BU entities were accepted by the clinical team and used in some cases to inform patient isolation status and timing. No $C_T$ values were available for outside tests, and they were only used if needed to document the time of a last negative or first positive test.

RT-qPCR primers targeting N1, N2, and Rnase P were used to evaluate each sample. Non-detectable (ND) $C_T$ values indicated that not enough viral RNA was present for amplification and detection and most often indicated the absence of viral RNA (negative infection). Samples with N1 and N2 target $C_T$ values above 40 or ND were considered SARS-CoV-2 negative. Samples with a $C_T$ under 40 for at least one target (N1 or N2) were considered a SARS-CoV-2 positive case and underwent further epidemiological and clinical follow-up. Our ruling of a positive test (only one N target positive) differs from the original CDC test protocol.

**Symptom attestation, surveys and contact tracing.** Any student, faculty, or staff visiting the university in-person for class, work, or research were required to submit daily symptom attestations and undergo routine testing. Daily symptom attestations were yes-no questionnaires regarding presence of the following: Fever of 100°F or feeling hot (if no thermometer available) accompanied by shivering/chills; new cough not related to chronic condition; difficulty breathing, shortness of breath; sore throat; new loss of taste or smell; vomiting; severe fatigue; severe muscle aches.

Individuals with positive symptom reports or status as a close contact to a confirmed BU positive case were followed by the contact tracing team, and if necessary, placed in quarantine or isolation. During the 10-day isolation mandated by Massachusetts DPH for positive cases, students were contacted every day and employees every other day for symptom presence or

progression; access to resources and mental health status were evaluated during each follow-up. Symptoms were recorded as dichotomous yes-no answers for the following categories: fever +/- chills, sore throat, cough, runny nose, difficulty breathing, shortness of breath, diarrhea, headache, fatigue, muscle aches, loss of smell and/or taste, foot sores/skin changes. Date of symptom onset was recorded in the patient's EMR.

**Data analysis.**   Microsoft Excel and R Studio were used to analyze collected data. Analysis of the raw $C_T$ values included all positive cases (n = 1633, **Table 1**). Cases were assigned a symptom classification of presymptomatic, symptomatic, or asymptomatic as defined above.

To make sure that there was no systematic change in average Rnase P values over the course of the study, the median and standard deviations of Rnase P values were compared across months (August 2020 to March 2021) via Kruskal-Wallis tests and Bartlett's test of homogeneity of variance to assess stability of the assay and consistency of the quality of the collected samples. The Rnase P target was used for quality control and was not used to normalize the N1 or N2 values.

Summary statistics of raw $C_T$ values for N1 and N2 qRT-PCR targets were calculated across asymptomatic, symptomatic, and presymptomatic groups. Non-parametric statistical tests, Kruskal-Wallis and Mann-Whitney U tests were used to assess any significant difference(s) in the $C_T$ values between symptom groups due to the non-normal distribution of $C_T$ values across the study population. Trends were analyzed for the entire dataset, across age brackets and for student versus employee populations.

Further, we addressed cases where only a single target amplified (N1 or N2) by comparing subpopulation with both targets to the subpopulations with only one target present.

**Table 1. Study population- demographics.**

|  | Total (n = 1633) | Students (n = 1207) | Employees (n = 426) |
|---|---|---|---|
| Age (years) (median, IQR) | 22 (20–29) | 21 (19–23) | 45 (32–55.8) |
| Sex, % female | 833 (51.0%) | 654 (54.2%) | 179 (42.0%) |
| Race and Ethnicity[a] |  |  |  |
| White | 804 (49.2%) | 553 (45.8%) | 251 (58.9%) |
| Hispanic/Latino | 238 (14.6%) | 170 (14.1%) | 68 (16.0%) |
| Asian | 170 (10.4%) | 143 (11.8%) | 27 (6.3%) |
| Black/African America | 106 (6.5%) | 51 (4.2%) | 55 (12.9%) |
| Two or more races [b] | 59 (3.6%) | 59 (4.9%) | -- |
| Native America, Native Hawaiian, or other Pacific Islander | 4 (0.3%) | 3 (0.2%) | 1 (0.2%) |
| Unknown [c] | 252 (15.4%) | 228 (18.9%) | 24 (5.6%) |
| Symptom Experience |  |  |  |
| Asymptomatic (n = 357, 21.9%) | 359 (22.0%) | 249 (20.6%) | 108 (25.3%) |
| Symptomatic (n = 521, 31.9%) | 520 (31.8%) | 452 (37.4%) | 69 (16.2%) |
| Presymptomatic (n = 755, 46.2%) | 754 (46.2%) | 506 (41.9%) | 249 (58.5%) |
| On-campus residential living | -- | 446 (37.0%) | -- |
| Employee Affiliation | -- | -- |  |
| Affiliate |  |  | 77 (18.1%) |
| Faculty |  |  | 45 (10.6%) |
| Staff |  |  | 304 (71.4%) |

[a]: Race and ethnicity were grouped in the electronic medical records from which this data was sourced, so it was only possible to report the two variables of race and ethnicity together for dataset totals to equal 100%.

[b]: This race/ethnicity code was only reported for students; thus, there was no available information on employees of two or more races.

[c]: This includes students with non-resident alien status

Comparisons of $C_T$ values across the groups of single- versus both-target amplification positives were conducted using Kruskal-Wallis and Mann-Whitney U tests.

**Ethics.** This study was classified as exempt from the need for informed consent from human subjects with BU's Charles River Campus Institutional Review Board.

## Results

### Study population (A)

Of the 1633 positive SARS-CoV-2 individuals included in this study, the median age was 22 years (interquartile range [IQR] 20–29), with students (n = 1207) being younger at 21 years (IQR 19–23) than employees (n = 426) at 45 years of age (IQR 32–55.8). There was a higher percentage of female students (n = 654/1207; 54.2%) than female employees (n = 179/426, 42.0%), though the total study population showed near equal binary sex distribution (n = 833/1633, 51.0% female). Close to half of the population reported White race (n = 804/1633, 49.2%), while the next largest race and ethnicity categories were Hispanic/Latino (238/1633, 14.6%), Asian (n = 170/1633, 10.4%), and Black/African American (n = 106/1633, 6.5%). Race and ethnicity breakdowns followed similar trends for students and employees, with the exception of more Black than Asian employees (12.9% versus 6.3%) and more Asian than Black students (11.8% versus 4.2%). Over one-third of students in the testing pool lived on the BU campus (n = 446/1207, 37.0%). Most employees were staff (n = 304/426, 71.4%) and the remainder were faculty members.

The greatest proportion of total cases were presymptomatic on the day of the positive test (n = 755/1633, 46.2%), followed by symptomatic at time of positive test (n = 521/1633, 31.9%) and asymptomatic for the infection course (n = 357/1633, 21.9%). Students and employees showed different distributions of symptom experience and onset. More students were symptomatic than remained asymptomatic (37.4% versus 20.6%), while fewer employees were symptomatic than remained asymptomatic. (16.2% versus 25.3%). For both students and employees, most infections were presymptomatic (41.9% and 58.5%, respectively), meaning that most positive individuals eventually experienced at least one symptom. During the study period, among students, there were no known hospitalizations due to COVID-19. Among employees, there were 6 known hospitalizations due to COVID-19.

### Rnase P analysis to assess assay consistency over time (B)

As a measure of the assay's performance over time, the monthly median Rnase P $C_T$ values associated with positive tests (n = 1633) were within a narrow range from 24.1 (IQR 21.8–27.4) to 26.2 (IQR 23.5–28.9), indicating consistency of the assay (**Fig 1** and **S1 Table**).

### General $C_T$ trends

For the total population, asymptomatic individuals showed the highest $C_T$ values for both N1 (median = 29.9, IQR 23.6–35.5) and N2 (median = 30.0, IQR 23.1–35.7). Presymptomatic individuals showed lower $C_T$ values than asymptomatic for N1 (median = 27.1, IQR 20.2–32.9) and N2 (median = 26.6, IQR 20.1–32.8). Symptomatic cases had the lowest $C_T$ values for N1 (median = 21.8, IQR 17.2–29.4) and N2 (median = 21.4, IQR 17.3–28.9). As lower $C_T$ values reflect higher viral load, symptomatic cases show the highest viral load, followed by presymptomatic cases, with asymptomatic cases showing the lowest viral load.

For the total population, N1 and N2 $C_T$ values were statistically different across the symptom classifications (presymptomatic, symptomatic, and asymptomatic) in a Kruskal-Wallis rank sum test (p< 0.001 for both N1 and N2). As seen in **Tables 2** and **S2** N1 and N2 $C_T$ values

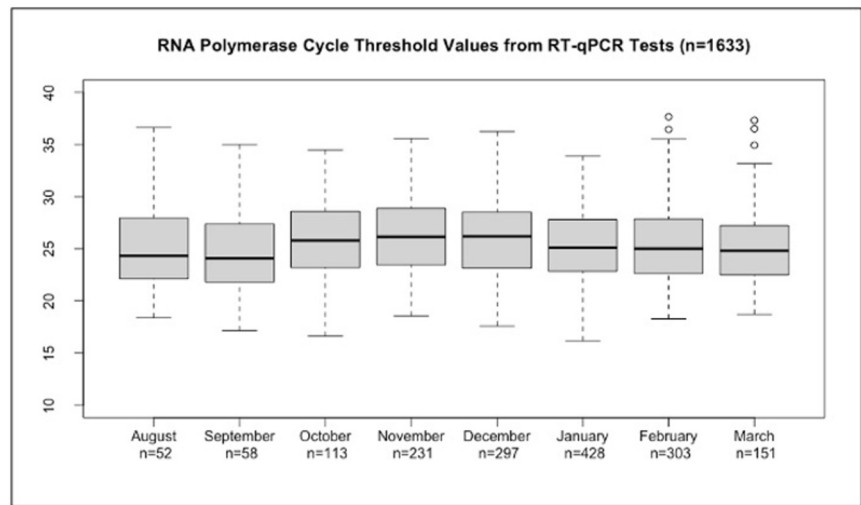

**Fig 1. RNA polymerase cycle threshold values by month over the course of the study.**

were highest in asymptomatic and lowest in symptomatic patients for all population subsets; these trends of $C_T$ values across asymptomatic, symptomatic, and presymptomatic cases are the same for each age group ($\leq 20$, 21–25, 26–30, $\geq 31$) and within the student and employee populations (all p<0.001) (**Table 3**).

## One vs. both N gene targets (D)

We defined a positive test as either both or one of the N1 or N2 targets being detectable. Not every positive test produced $C_T$ values for both N1 and N2 targets. This method is different than the CDC protocol, which requires both N1 and N2 results to be positive for the overall test result to be declared positive. A consequence is that in theory our testing strategy will be

**Table 2. General $C_T$ trends for the N1 gene target.**

| Whole data set with N1 amplified (n = 1557) | Total | Asymptomatic[a] (n = 319) | Symptomatic[b] (n = 512) | Presymptomatic[c] (n = 726) |
|---|---|---|---|---|
| **Total** (median, (Q1- Q3)) | 25.9 (19.5–32.8) | 29.9* (23.6–35.5) | 21.8* (17.2–29.4) | 27.1* (20.2–32.9) |
| **By Age** (median, (Q1- Q3)) | | | | |
| $\leq$20 years (n = 498) | 25.9 (19.3–33.0) | 29.7* (22.2–35.5) | 21.8* (16.7–30.4) | 27.8* (21.0–33.3) |
| 21–25 years (n = 537) | 26.2 (19.9–33.3) | 31.6* (26.0–35.8) | 21.4* (17.5–29.0) | 28.1* (20.6–33.5) |
| 26–30 years (n = 161) | 26.1 (18.8–31.9) | 30.3* (26.5–36.7) | 21.8* (17.7–28.3) | 28.5* (20.2–32.1) |
| $\geq$31 years (n = 361) | 25.3 (19.5–31.4) | 28. 3* (22.5–34.6) | 22.7* (18.8–29.0) | 25.0* (19.5–30.6) |
| **By School Affiliation** (median, (Q1- Q3)) | | | | |
| | Total | Asymptomatic (n = 222) | Symptomatic (n = 445) | Presymptomatic (n = 486) |
| **Students** (n = 1153) | 26.1 (19.4–33.2) | 30.3* (23.8–35.7) | 21.8* (17.3–30.3) | 28.0* (20.8–33.4) |
| | Total | Asymptomatic (n = 97) | Symptomatic (n = 67) | Presymptomatic (n = 240) |
| **Employees** (n = 404) | 25.3 (19.7–31.4) | 28.3* (23.4–35.0) | 21.3* (16.4–25.6) | 25.3* (19.5–31.0) |

[a] Individuals who did not experience any of the monitored symptoms over infection course.

[b] Individuals who were experiencing symptoms before testing positive.

[c] Individuals who developed symptoms the day of or days after positive test.

* p-value is <0.001 using a Kruskal-Wallis rank sum test to compare $C_T$ values across symptom categories of asymptomatic, symptomatic, and presymptomatic for each variable; alpha = 0.05 used to assess any significant difference between median $C_T$ values.

**Table 3. N1 $C_T$ values by age group.**

| Age Groups (years) | N1 (median, IQR (Q1-Q3)) | | | | p-value[a] |
|---|---|---|---|---|---|
| | ≤20 | 21–25 | 26–30 | ≥31 | |
| Asymptomatic | 29.7 (22.2–35.5) | 31.6 (26.0–35.8) | 30.3 (26.5–36.7) | 28.3 (22.5–34.6) | 0.20 |
| Symptomatic | 21.8 (16.7–30.4) | 21.4 (17.5–29.0) | 21.8 (17.7–28.3) | 22.7 (18.8–29.0) | 0.94 |
| Presymptomatic | 27.8 (21.0–33.3) | 28.1 (20.6–33.5) | 28.5 (20.2–32.1) | 25.0 (19.5–30.6) | 0.029* |

[a] Kruskal-Wallis rank sum tests were conducted for the three symptom groups to assess any difference between Ct values across age categories.

* significant at $p \leq 0.05$

somewhat more sensitive than the CDC definition, while somewhat less specific, though this conclusion rests on unexamined assumptions about the specificity of a one vs. two target strategies. While most cases produced two amplified targets (n = 1432/1633, 87.7%), 7.7% of cases only amplified N1 (n = 125/1633), and 4.7% only amplified N2 (n = 76/1633) (**Fig 2**). This is reassuring insofar as our results would only differ slightly had we been using the CDC definition. Most of the population with both targets detected were presymptomatic (48.0%), followed by symptomatic (34.2%) and then asymptomatic (17.7%). For the population with only one target detected, most cases were asymptomatic (52.0% for N1-Only, 50.0% for N2-Only), while a minority of cases were symptomatic (17.6% for N1-Only, 11.8% for N2-Only) (**Table 4**). There was no significant difference in the distribution of asymptomatic, symptomatic, or presymptomatic individuals between the N1-only and N2 only populations (p = 0.39).

Patients with detectable values for both targets had significantly lower N1 $C_T$ values (median = 24.8, IQR 19.2–31.6) than those with only N1 detectable (median = 36.2, IQR 34.9–37.3) (p<0.001). Similarly, patients with both targets had significantly lower N2 values (24.9, IQR 19.2–31.7) than those with only N2 (37.0, IQR 36.2–37.9) (p<0.001), which follows the trend of asymptomatic individuals having the highest $C_T$ values. Within these populations, those with both targets amplified showed significant differences in $C_T$ values between asymptomatic, symptomatic, or presymptomatic groups for N1 and N2 (p<0.001) (**Fig 3**). However, those with only N1 or N2 amplified did not show a difference in $C_T$ values between asymptomatic, symptomatic, or presymptomatic groups for N1 or N2 $C_T$ values (**S2 Table**).

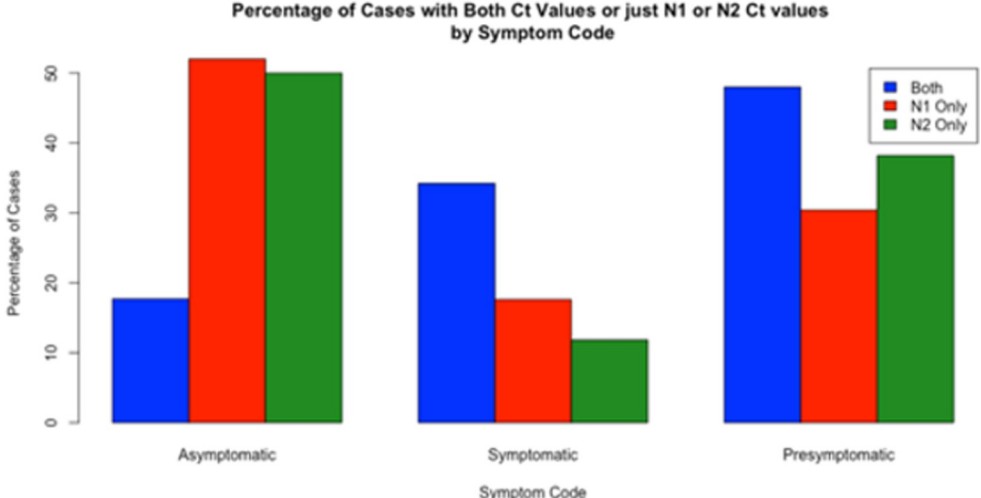

**Fig 2. Percentage of cases with either both or single target(s) amplified by symptom classification.**

**Table 4. Amplification of one vs. both targets.**

|  | Both (n, %) | N1 Only (n, %) | N2 Only (n, %) |
|---|---|---|---|
| Total (n = 1633) | 1432 (87.7) | 125 (7.7) | 76 (4.7) |
| Asymptomatic | 254 (17.7) | 65 (52.0) | 38 (50.0) |
| Symptomatic | 490 (34.2) | 22 (17.6) | 9 (11.8) |
| Presymptomatic | 688 (48.0) | 38 (30.4) | 29 (38.2) |

### Symptom type, load, and trends (E)

Within the entire population, 78.1% experienced at least one symptom during surveillance by the clinical team (n = 1276/1633). The mean number of symptoms per symptomatic and presymptomatic individual is stable over the course of the study. Of those ever experiencing symptoms, the most common symptoms were nasal congestion (73%, n = 932, 1276), cough (60.0%, n = 761/1276), fatigue (59.0%, n = 753/1276), and sore throat (53.1%, n = 678/1276). The least common symptoms were diarrhea (12.5%, n = 160/1276), dyspnea on exertion (DOE) (6.9%, n = 88/1276), foot or skin changes (including rash) (4.2%, n = 53/1276), and vomiting (2.1%, n = 27/1276) (**Tables 5 and S3 and S1 Fig**).

## Discussion

In this analysis, we detected a strong relationship between RT-qPCR signal intensity and the presence or absence of symptoms. Clear trends showed asymptomatic individuals had the lowest viral loads (highest $C_T$ values), and symptomatic individuals had the highest viral loads, while presymptomatic individuals fell in between these extremes. If we assume that detection of only one of the N1 or N2 targets is a further reflection of waning signal intensity, then these proportions are again reflected in that the highest rate of single target detections was in those who were asymptomatic. Extrapolating further, our results support the theory that symptomatic individuals have, on average, higher viral loads than those who are pre- or asymptomatic. These trends have been observed elsewhere [12].

Only 12.4% of the positive tests had a single detectable $C_T$ value (N1 or N2). It is important to note that many RT-qPCR tests used under EUA during the pandemic adjudicated their

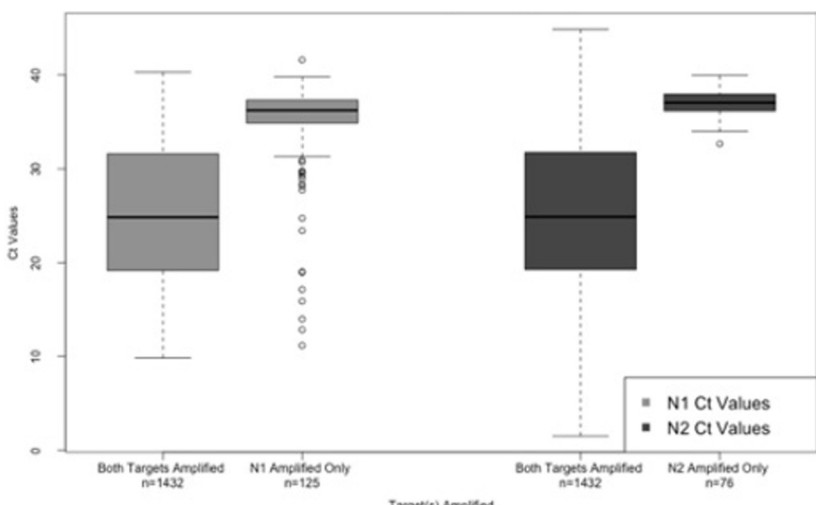

**Fig 3. Comparison of N1 and N2 cycle threshold values for tests with both or only one target amplified.**

**Table 5. Proportion of population experiencing symptoms[a] by symptom type.**

| Symptom Type[b] | Symptom-experiencing population[c] (n = 1276) |
|---|---|
| Nasal Congestion | 73.0% (932/1276) |
| Cough | 60.0% (761/1276) |
| Fatigue | 59.0% (753/1276) |
| Sore Throat | 53.1% (678/1276) |
| Loss of Smell or taste | 50.3% (642/1276) |
| Headache | 48.7% (621/1276) |
| Muscle Ache | 44.0% (561/1276) |
| Fever +/- Chills | 40.0% (506/1276) |
| Nausea | 13.6% (173/1276) |
| Shortness of Breath | 13.5% (172/1276) |
| Diarrhea | 12.5% (160/1276) |
| Dyspnea on Exertion | 6.9% (88/1276) |
| Foot or Skin Changes/Rash | 4.2% (53/1276) |
| Vomiting | 2.1% (27/1276) |

[a] Asymptomatic patients were excluded from this table.

[b] These 14 symptoms were monitored by the BU Contact Tracers

[c] Percentage of individuals who did experience each symptom during follow-up period are reported.

results requiring that both targets be detectable. Our approach favored sensitivity over specificity, since our working hypothesis was that detecting asymptomatic individuals with even low viral loads would more effectively limit spread on a densely packed urban campus. This hypothesis is countered by the argument that tests with a single undetectable $C_T$ value are more likely to be from cases at the end of the disease course, or those with very low viral loads throughout the course of disease, both of which are less likely to spread disease to others. The majority of those in this study with only one amplified $C_T$ were indeed asymptomatic. Future work will look at whether $C_T$ values predict the ability of an individual to infect others.

Overall, most patients in this study experienced at least one symptom at some time either before or within 10 days of testing positive. Because symptoms were self-reported by patients and thus represent some subjectivity in experience (including the seasonality of allergies and other non-COVID illnesses like flu), some random misclassification of symptoms may have been introduced. With that said, the reporting of symptoms was independent of PCR results, and therefore should not introduce selection bias. Moreover, every interview for symptom reporting was conducted by a trained health care professional and judged as a dichotomous yes-no variable. It is worth noting that during this time period, all positive individuals regardless of symptoms were isolated for the same amount of time, thus reducing the incentive to deny the presence of symptoms. In addition, clinical staff did not in general have access to $C_T$ values.

Further work should address trends of symptom duration and symptom severity, neither of which were analyzed in this study, and how these relationships may be altered following Covid vaccinations. Nearly all the data for this analysis were generated prior to vaccine licensure, and only a minority of individuals, all sampled at the end of the observation period, had been vaccinated. Other limitations include the absence of vaccination status as a variable in this analysis. Only 5.5% of the population with a positive test from August 8, 2020, to March 18, 2021 reported vaccinations (n = 90/1633) at time of data collection. Additionally, BU did not require reporting vaccination status in the spring of 2021, which further prevented this analysis from commenting on the interaction of vaccination status with cycle threshold values or symptom experience.

Sequence data were not incorporated in this study, so the authors are unable to comment regarding variants of concern (VOC), including the later-to-appear Delta and Omicron variants, or the effect of vaccination on VOCs and viral load. Sequencing initiatives begun at BU in January 2021 established that most of the local spread was still wild type virus, with the Alpha variant emerging over the January–March 2021 timeframe in the local area.

In conclusion, consistent with our hypothesis, PCR signal intensity was strongly associated with symptomatology. Those who presented with symptoms at the time of diagnosis had the lowest $C_T$ values, while those who remained asymptomatic throughout had the highest $C_T$ values. Because PCR signal intensity is a measure of viral load, and by extrapolation likely a measure of infectiousness, our data support the theory that asymptomatic patients are generally less infectious than symptomatic patients.

## Supporting information

**S1 Table. RNA polymerase cycle threshold values by month over the course of the study.**
(DOCX)

**S2 Table. General $C_T$ trends of N2 target.**
(DOCX)

**S3 Table. N2 $C_T$ values by age group.**
(DOCX)

**S4 Table. Comparison of N1 and N2 cycle threshold values for tests with both or only one target amplified.**
(DOCX)

**S5 Table. Symptom type broken out by classification group and in the student and employee population.**
(DOCX)

**S1 Fig. Symptom analysis.**
(TIF)

## Author Contributions

**Conceptualization:** Christopher J. Gill, Lynn Doucette-Stamm, Laura White, Davidson H. Hamer, Catherine M. Klapperich.

**Data curation:** Samantha M. Hall, Lena Landaverde, Grace M. Yee, Madison Sullivan, Lynn Doucette-Stamm, Hannah Landsberg, Judy T. Platt, Laura White, Davidson H. Hamer, Catherine M. Klapperich.

**Formal analysis:** Samantha M. Hall, Laura White, Davidson H. Hamer, Catherine M. Klapperich.

**Investigation:** Samantha M. Hall, Catherine M. Klapperich.

**Methodology:** Samantha M. Hall, Christopher J. Gill, Grace M. Yee, Lynn Doucette-Stamm, Hannah Landsberg, Judy T. Platt, Laura White, Davidson H. Hamer, Catherine M. Klapperich.

**Project administration:** Lena Landaverde, Grace M. Yee, Madison Sullivan, Judy T. Platt, Catherine M. Klapperich.

**Software:** Laura White.

**Supervision:** Lynn Doucette-Stamm, Hannah Landsberg, Judy T. Platt, Davidson H. Hamer.

**Validation:** Laura White.

**Writing – original draft:** Samantha M. Hall, Lena Landaverde, Christopher J. Gill, Laura White, Davidson H. Hamer, Catherine M. Klapperich.

**Writing – review & editing:** Samantha M. Hall, Lena Landaverde, Christopher J. Gill, Madison Sullivan, Lynn Doucette-Stamm, Hannah Landsberg, Judy T. Platt, Laura White, Davidson H. Hamer, Catherine M. Klapperich.

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
