## [Decision Letter · Decision Letter 0]

1 Mar 2022

PONE-D-22-02578Comparison of Anterior Nares Viral Loads in Asymptomatic and Symptomatic Individuals Diagnosed with SARS-CoV-2 in a University Screening ProgramPLOS ONE

Dear Dr. Klapperich,

Thank you for submitting your manuscript to PLOS ONE. After careful consideration, we feel that it has merit but does not fully meet PLOS ONE’s publication criteria as it currently stands. Therefore, we invite you to submit a revised version of the manuscript that addresses the points raised during the review process. The academic editor has to agree with the comments from the reviewer:  The results for the N1 and N2 genes were so similar, and thus it is necessary to show them separately for every comparison. 

We look forward to receiving your revised manuscript.

Kind regards,

Etsuro Ito

Academic Editor

PLOS ONE

Journal Requirements:

[I have read the journal's policy and the authors of this manuscript have the following competing interests: Dr. Klapperich is a co-founder of Biosens8, Inc.] 

5. Please ensure that you refer to Figure 2 in your text as, if accepted, production will need this reference to link the reader to the figure.

Reviewers' comments:

Reviewer's Responses to Questions

**Comments to the Author**

1. Is the manuscript technically sound, and do the data support the conclusions?

Reviewer #1: Yes

2. Has the statistical analysis been performed appropriately and rigorously? 

Reviewer #1: Yes

3. Have the authors made all data underlying the findings in their manuscript fully available?

Reviewer #1: Yes

4. Is the manuscript presented in an intelligible fashion and written in standard English?

Reviewer #1: Yes

5. Review Comments to the Author

Reviewer #1: This is a valuable paper with results that will be of interest to decision makers interpreting anterior nares PCR results for COVID-19 testing. Overall, I found the paper to be well written and the study approach and findings to be clearly explained.

In addition to the specific comments below, I would offer the following general comments. The results for the N1 and N2 genes were so similar, I wonder if it is necessary to show them separately for every comparison. The information on results with only one of the two genes detected was interesting, but otherwise, it seems like you combine them in some way (e.g. take midpoint) or just show the analysis for one gene and say the other led to the same conclusions.

The authors could reduce the number of significant digits throughout the paper. Though common to show tenths for percentages, it does tend to imply more precision than is warranted and makes the text and tables harder to read. Also consider the same comment for p-values, e.g. Table 3 don’t need to be shown to 4 digits.

Finding employees were more likely to report no symptoms than students was an interesting result in light of the conventional wisdom that older age groups tend to be more severely affected. Please comment on the discussion on whether this might be due to testing frequency or other factors and compare your results to analogous comparisons from other studies if available.

Title: Because viral loads were not measure directly, I would recommend changing to “Comparison of Anterior Nares Ct Values in Asymptomatic …..”

Abstract: “gold standard method” PCR based on various sample types is certainly the most widespread testing method. I hesitate a bit to call it the gold standard. While it is very sensitive for detecting current or past presence of virus, it is not specific for detection of the infectious period. Also, the performance of PCR of course depends on sample source.

We collected the N1 and N2 Ct values… I would say collected N1 and N2 target gene Ct values in the first usage

“n=931, 1276” should be 932/1276

Introduction: consider use of gold standard as noted above.

Methods: “determined to be a false positive by the clinical team” What were the criteria for that determination?

“Symptomatic patients showed symptoms before the day of testing” I found it unclear whether symptoms prior to testing were self-reported contemporaneously through a questionnaire or determined retrospectively through the clinical interview.

Results: For discussion, wow did demographic data of the study population compare with the student body and employee demographics?

“with asymptomatic cases showing the lowest viral load as expected” suggest deleting “as expected” when presenting results.

Define “staff affiliates”

Remined >> remained

Were any hospitalizations reported?

Discussion:

“every interview for symptom reporting was conducted by a trained health professional” See note on pretesting symptom reporting above. Also, did interviewers or the respondents have access to Ct values prior to the interview?

Figures and Tables

Need to define parenthetic ranges in the tables.

Table 3: It is not clear which comparisons the p-values apply to.

6. PLOS authors have the option to publish the peer review history of their article (what does this mean?). If published, this will include your full peer review and any attached files.

Reviewer #1: No

---

## [Author Response · Author response to Decision Letter 0]

10 Jun 2022

All comments are addressed in the attached cover letter and response to reviewers document.

---

## [Editor Report · Decision Letter 1]

16 Jun 2022

Comparison of Anterior Nares CT Values in Asymptomatic and Symptomatic Individuals Diagnosed with SARS-CoV-2 in a University Screening Program

PONE-D-22-02578R1

Dear Dr. Klapperich,

We’re pleased to inform you that your manuscript has been judged scientifically suitable for publication and will be formally accepted for publication once it meets all outstanding technical requirements.

Kind regards,

Etsuro Ito

Academic Editor

PLOS ONE

---

## [Editor Report · Acceptance letter]

4 Jul 2022

PONE-D-22-02578R1 

Comparison of Anterior Nares CT Values in Asymptomatic and Symptomatic Individuals Diagnosed with SARS-CoV-2 in a University Screening Program 

Dear Dr. Klapperich:

I'm pleased to inform you that your manuscript has been deemed suitable for publication in PLOS ONE. Congratulations! Your manuscript is now with our production department. 

Kind regards, 

on behalf of

Prof. Etsuro Ito 

Academic Editor

PLOS ONE